# Adaptation Response in Sheep: Ewes in Different Cortisol Clusters Reveal Changes in the Expression of Salivary miRNAs

**DOI:** 10.3390/ani13203273

**Published:** 2023-10-20

**Authors:** Isabella Manenti, Irene Viola, Ugo Ala, Paolo Cornale, Elisabetta Macchi, Paola Toschi, Eugenio Martignani, Mario Baratta, Silvia Miretti

**Affiliations:** 1Department of Veterinary Sciences, University of Torino, Largo Paolo Braccini 2, 10095 Grugliasco, Italy; isabella.manenti@unito.it (I.M.); irene.viola@unito.it (I.V.); ugo.ala@unito.it (U.A.); elisabetta.macchi@unito.it (E.M.); paola.toschi@unito.it (P.T.); eugenio.martignani@unito.it (E.M.); 2Department of Agricultural, Forestry and Food Sciences (DISAFA), Animal Production Unit, Largo Paolo Braccini 2, 10095 Grugliasco, Italy; paolo.cornale@unito.it; 3Department of Chemistry, Life Sciences and Environmental Sustainability, University of Parma, Parco Area delle Scienze 11a, 43124 Parma, Italy; mario.baratta@unipr.it

**Keywords:** animal welfare, circulating miRNA, stress response, sheep, saliva

## Abstract

**Simple Summary:**

Monitoring animals’ welfare is a tricky challenge. The responsiveness to stressors is subjective and its magnitude varies among species, breeds, and individuals. Small molecules named miRNAs obtained from body fluid samples are promising biomarkers to classify animal welfare. Under common farming conditions, we identified several miRNAs in the saliva of ewes. Based on the concentration and trend of individual salivary cortisol, one of the molecules historically recognized and used to define the stress level in vertebrates, we divided the animals into two clusters. The expression of two miRNAs involved in regulating genes related to the stress pathway resulted in significant differences between the two phenotypes, and their trends were correlated. These results have produced objective data to improve the knowledge about sheep physiology and the basis to develop new tools for welfare assessment. In the animal production field, the assessment of biomarkers that are useful in identifying individuals with different levels of vulnerability or resiliency to stress stimuli may help to introduce new strategies in animal farms and genetic research for the selection of animals.

**Abstract:**

Farm procedures have an impact on animal welfare by activating the hypothalamic-pituitary-adrenal axis that induces a wide array of physiological responses. This adaptive system guarantees that the animal copes with environmental variations and it induces metabolic and molecular changes that can be quantified. MicroRNAs (miRNAs) play a key role in the regulation of homeostasis and emerging evidence has identified circulating miRNAs as promising biomarkers of stress-related disorders in animals. Based on a clustering analysis of salivary cortisol trends and levels, 20 ewes were classified into two different clusters. The introduction of a ram in the flock was identified as a common farm practice and reference time point to collect saliva samples. Sixteen miRNAs related to the adaptation response were selected. Among them, miR-16b, miR-21, miR-24, miR-26a, miR-27a, miR-99a, and miR-223 were amplified in saliva samples. Cluster 1 was characterized by a lower expression of miR-16b and miR-21 compared with Cluster 2 (*p* < 0.05). This study identified for the first time several miRNAs expressed in sheep saliva, pointing out significant differences in the expression patterns between the cortisol clusters. In addition, the trend analyses of these miRNAs resulted in clusters (*p* = 0.017), suggesting the possible cooperation of miR-16b and -21 in the integrated stress responses, as already demonstrated in other species as well. Other research to define the role of these miRNAs is needed, but the evaluation of the salivary miRNAs could support the selection of ewes for different profiles of response to sources of stressors common in the farm scenario.

## 1. Introduction

Commonly, the term “stress” is perceived with a negative connotation, but scientifically, it is an adaptation response of the organism to stimuli, which alters its homeostatic balance. Physiological nonspecific responses are induced by several stimuli, and the hypothalamic-pituitary-adrenal (HPA) axis is primarily involved in preserving homeostasis and allowing animals to cope with endogenous factors and environmental changes [1]. This mechanism has an essential impact on the health and productivity of these animals [2].

Keeping farm animals clinically healthy and without distress (i.e., severe and/or prolonged stressful status) is fundamental to producing safe and high-quality food, but assessing animals’ welfare is challenging. This topic is highly relevant for governments, food industries, and consumers whose choices are increasingly influenced by extrinsic factors like product origin, general production practices, and animal welfare.

Farm procedures and management play a crucial role in animal welfare and productivity, inducing metabolic and molecular changes that can be monitored on different matrices by the quantitative and qualitative measurement of markers [3,4].

Usually, glucocorticoids have been assayed to assess the response to stress and are considered indicators of the welfare state. Often, they are measured in plasma and serum. Still, it is considered preferable to measure them in other biological matrices including saliva, urine, feces, and hair because these matrices can be sampled without the risk of inducing a glucocorticoid response [5]. Moreover, the techniques of sampling and analysis available are feasible and practical for measuring to be performed both at an individual and group level [6].

In sheep saliva, classical stress markers comprise endocrine and chemical secretion changes, especially in the levels of hormones or enzymes such as cortisol and α-amylase, respectively [7,8].

The main difficulty in defining the state of welfare is due to the individuality of the adaptation response, which varies among individuals and populations [9,10] because it reflects genetic predisposition and experiences affecting epigenetic factors [11]. Indeed, stress is defined as an individual’s cognitive, emotional, behavioral, and physiological response to endogenous or exogenous stimuli [12].

MicroRNAs (miRNAs) are small non-coding RNAs (~22 nucleotides) that play a role in the post-transcriptional regulation of several cellular processes including development, immune responses, and homeostasis. Expressed by tissue, they are released in body fluids and are therefore called circulating miRNAs (c-miRNAs) [13]. Changing their profile under different conditions like physiological, pathological, and psychological states can be used as minimally invasive biomarkers [14].

In the past few years, several studies have demonstrated in humans and rodents that the expression of miRNAs in the brain changes in response to alterations in the external environment, and it has been associated with experiential states such as stress, depression, and anxiety [15]. The miRNA expression changes are not only at the neuronal tissue level, but were found to be highly correlated with variation in the circulating peripheral fluids in patients affected by psychological disorders [16]. Emerging evidence identifies these molecules as biomarkers of resilience or vulnerability to stress. In particular, c-miRNAs are considered among the most promising biomarkers of stress-related disorders in humans and animals [17], but at present, studies on larger animals such as farm animals are still lacking.

The present study aimed to determine the salivary miRNA profiles associated with variable stress responses in ewes following a common farm management practice: the introduction of a ram in the flock. In detail, the purposes of this research were to (a) identify the salivary cortisol level to classify the responsiveness of animals, (b) explore and define a panel of expression of c-miRNAs in the saliva samples collected from the Frabosana-Roaschina breed, and (c) check for possible correlations to evaluate the predictive value of salivary miRNAs for different cortisol phenotypes.

## 2. Materials and Methods

The study is framed within the SmartSheep project. Study approval was provided by the Italian Ministry of Health, authorization number 494/2020-PR. All methods were carried out following relevant guidelines and regulations. Informed consent is not required.

Figure 1 shows the experimental design and analysis flowchart of the study.

### 2.1. Ewes

For the study, 20 ewes of the same age (10–12 months old) and Frabosana-Roaschina breed, weighing approximately 52 ± 4.3 kg, were considered. During the experimental period of 43 days, all ewes were kept in the sheepfold of the Veterinary Department of Torino University and managed according to the breeder’s indications. During the first 30 days, the animals were given time to adapt to the new environment. The sample period began 6 days before a ram of the same breed belonging to the same farm was introduced into the flock.

### 2.2. Saliva Collection

Saliva was sampled on alternate days, for a total of 6 samples for each ewe, 3 before (PRE) and 3 after (POST) the introduction of the ram (Figure 1a). Saliva samples were collected early in the morning, before feeding time, using Salivette^®^ with a polyethylene pad (Sarsted AG & Co., Nümbrecht, Germany). The swab was inserted into the animals’ mouths with a clamp for half a minute to allow the sheep to chew it, thus absorbing the saliva. The saliva was collected from the swab after sampling by centrifugation at 3500 rpm for 10 min at 5 °C and stored at −20 °C until processing [18,19,20].

### 2.3. Salivary Cortisol Estimation

Salivary cortisol was determined for all the saliva samples of the 20 ewes (n = 120) to detect the concentration of the hormone during the experimental trial. For this purpose, an enzyme immunoassay has been performed with a commercial multi-species EIA Cortisol Kit (K003 Arbor Assays^®^, Ann Arbor, MI, USA) validated for saliva and other biological substrates in sheep, according to the manufacturer’s protocol [6]. All analyses were repeated twice. The inter and intra-assay coefficients of variation were less than 10% (7 and 9%, respectively). The test’s sensitivity was determined by measuring the least amount of hormone standard consistently distinguishable from the zero concentration of the standard, and was calculated to be 25.4 pg/mL. Serial dilutions (1:4, 1:8, 1:16, and 1:32) of the saliva samples from six sheep were assayed to test for parallelism against the standard curve (r^2^ = 0.981). The mean recovery rate of the cortisol added to the saliva samples was 95.8% (n = 6). According to the manufacturer, the kit displayed the following cross-reactivities: 100% with cortisol, 18.98% with dexamethasone, 7.8% with prednisolone, and 1.2 with corticosterone. The results are expressed as the amount of cortisol in saliva (ng/mL).

### 2.4. Extraction of Salivary miRNAs

For the extraction of c-miRNAs, only clear saliva samples were considered (n = 114). Indeed, the results obtained from greenish saliva are distorted compared to those obtained from clean saliva.

Before starting extraction, the samples were centrifugated at 15,000 rpm for 5 min at +4 °C temperature. Extraction was performed starting from 500 μL of centrifugated saliva with the Maxwell^®^ RSC miRNA Plasma or Serum kit (Promega, Madison, WI, USA), according to the manufacturer’s protocol. During extraction, 1 μL of UniSp2,4,5 spike-in (Qiagen, Hilden, Germany) was added to each sample to check the quality of extraction.

MiRNA quantification was performed using the Qubit™ microRNA Assay kit together with the Quantus™ Fluorometer (Invitrogen, Thermo Fisher Scientific Inc, Waltham, MA, USA), following guidelines.

The miRNA samples were then stored at −80 °C.

### 2.5. cDNA Synthesis and Quantitative Real-Time PCR

Immediately after extraction and quantification, 0.8μL of the microRNA samples were retrotranscribed in cDNA using the miRCURY LNA™ RT Kit (Qiagen, Hilden, Germany) in a final volume of 10 μL, according to the manufacturer’s instructions. At this stage, 0.5 μL of UniSp6 spike-in (Qiagen, Hilden, Germany) was added to each sample for the evaluation of the quality of the reverse transcription process.

The cDNA samples were stored at −20 °C until processing.

The cDNA was diluted 30 folds before use and 3 μL was assayed with 7 μL of PCR mix, according to the protocol for the miRCURY LNA™ SYBR Green PCR Kit (Qiagen, Hilden, Germany). The PCR amplification was performed on the Bio-Rad CFX Connect Real-Time System (Bio-Rad, Hercules, CA, USA). Three technical replicates were performed for each biological sample, and the average Cq values of each triplicate were calculated.

### 2.6. Statistical Analyses

Clustering analysis based on the normalized cortisol values has the main purpose of highlighting subsets of animals that share a homogeneous range of values and simultaneously also a common expression trend. Since we did not have an a priori hypothesis as to what number of groups the sample could be divided into, we opted for hierarchical clustering so that we could choose *a posteriori* where to cut into clusters and thus determine the number of clusters to be considered. While looking for clusters homogeneous in sample size, we chose to identify two. Specifically, sample clustering analyses have been conducted in R (version 4.3.1 (16 June 2023)—”Beagle Scouts”) and Rstudio (1 June 2023 Build 524). The distances between samples were calculated with both the Euclidean and correlation functions, the latest by using the “pairwise.complete.obs” parameter. The hierarchical clustering has been conducted by the “hclust” function with the method parameter set to “complete”. The similarities between clusters were evaluated by the hypergeometric test (phyper function in R).

For statistical analyses of miRNAs’ differential expression, the Mann–Whitney U test in Software IBM^®^ SPSS^®^ statistic for Windows, version 26.0, Armonk, NY 2019 has been used. Significance has been fixed for *p* < 0.05. The statistical analyses were carried out on ∆Ct, obtained from the difference between the Ct of each sample and the Ct of the normalizer Unisp6. For this reason, the higher the level of the ∆Ct, the less the miRNA is expressed.

## 3. Results

### 3.1. Salivary Cortisol

Salivary cortisol levels range between 0.08 and 6.7 ng/mL and have an average value of 1.18 (and SD, ±0.41) ng/mL. The clustering analysis was conducted based on the normalized cortisol values in order to group samples according to the value magnitude with Euclidean function, and to the trends of these values, with correlation function (Figure 2). By overlapping the similar subgroups obtained with the two measures, animals were divided into two clusters (Cluster 1:10 animals and Cluster 2:5 animals; Table 1), before being taken into consideration for the analyses. Five animals were not considered for the following analysis of the miRNAs. In detail, the animal with the ID number 48 clustered alone, while the animals with ID numbers 54, 57, 62, and 65 did not show cluster similarities in both analyses and, consequently, were not representative of a single group.

### 3.2. miRNAs Expression

In total, 16 miRNAs expressed in saliva and related to stress (Table 2) were selected from the literature [21,22,23] and tested on a representative number of samples (n = 30, 1 PRE and 1 POST samples for each ewe belonging to cluster 1 and 2). However, 7 miRNAs (miR-16b; -21; -24; -26a; -27a; -99a, and -223) detected reliably in the analysis were tested on all samples (n = 87). Among these, miR-16b and miR-21 showed significantly less expression in animals in Cluster 1 (*p* < 0.05) (Figure 3). Data were expressed with the function 2^−∆Ct^, which was used to calculate the relative expression of the target miRNAs [24].

For miR-16b, the minimum value of ∆Ct is 1.478 and the maximum is 2.256, with a median value of 1.909. The median of the ∆Ct value in Cluster 2 is 1.826, and in Cluster 1 is 1.971. For miR-21, the minimum value of the ∆Ct value is 0.776 and the maximum is 2.029, with a median value of 1.596. The median of the ∆Ct value in Cluster 2 is 1.486, and in Cluster 1 is 1.651.

### 3.3. miRNAs’ Expression Clustering

Since mir-16b and mir-21 were differentially expressed, we evaluated whether the expression profiles of these miRNAs are also able to further characterize the 15 subjects analyzed and, in particular, to subdivide them into similar subgroups with common subjects. The animals were clustered by applying a hierarchical clustering based on a correlation distance on the two miRNAs’ expression profiles, respectively (Figure 4), and two clusters were highlighted according to both miRNAs’ expression values. The overlap of the most similar clusters based on miR-16b and miR-21 is significant (*p* = 0.017), delineating a similar pattern of miRNA expression between the animals (50, 53, 58, 59, 60, 63 in one cluster; 46, 52, 55, 56, 61, 64 in the other cluster). Correlation analyses between the miRNAs’ expression profiles and cortisol levels were performed. No significance has emerged from these analyses.

## 4. Discussion

Stress and welfare are complex to define because the perception of the stress stimuli and the consequent adaptation response is individual [12].

Besides ethical-related issues, the impact of stress and consequent animal welfare status has a direct repercussion on production in terms of reproduction efficiency, meat and milk quality, and quantity; therefore, it is of major economic relevance for the food industry.

Farming management practices can be a source of stressors and can trigger the body’s response, inducing the activation of the HPA axis with a consequent individual rate of gain in corticosteroid levels [25]. Recently, studies have increasingly focused on individual differences in stress responses, and they have revealed considerable differences in concentration levels and trends of stress-related molecules among animals [26,27]. Our choice to use the introduction of a ram in the flock of ewes as a stimulus arises from the volute to retrace a common procedure adopted in this breeding system. So far, it is established that the ram induces effects in ewes through the activation of the hypothalamo-pituitary gonadal axis that is closely related to the HPA axis, producing an alteration in socio-sexual interaction [28]. Moreover, McCosh and colleagues have demonstrated that exposing ewes to rams during the transition into the breeding season caused an increase in blood cortisol pulse duration [29]. Here, we have focused on the differential expression of salivary miRNAs in response to this practice. However, after comparing the average cortisol concentrations before and after exposure to the ram, no significant alterations were found.

We have selected saliva as a biological matrix preferable to blood because in individuals without training, it can be collected easily and in a non-invasive way with a lower stress for the animals. Saliva was used in previous studies to evaluate stress in sheep [6,30,31] and, in some research, it has been used to evaluate miRNAs as biomarkers of stress in other species such as humans [32], pigs [33], and mice [34].

MiRNAs play an important role in physiological and psychological functioning at a cellular and genetic level in mammal organisms. Evidence shows that miRNAs are excreted physiologically by brain tissue in response to stress, and this point assigns them a potential biomarker role for stress-related disorders [35]. It appears that the research of free c-miRNAs in saliva samples could be a strength due to this matrix, and the potential of salivary miRNAs has already been tested in an exploratory analysis of human social stress [19]. Moreover, the collection, processing, storage, and detection of samples are relatively cost-effective in standard laboratory conditions [36].

To the authors’ knowledge, no studies on circulating miRNAs in the saliva of sheep have been published. Here, we extracted and analysed for the first time c-miRNAs in sheep saliva and, furthermore, we observed differences in miRNAs’ expression between animals of the same age, breed, and flock, but different for salivary cortisol clustering. Our results are in line with a previous study by Wiegand and colleagues that demonstrated a significant difference in expression for miR-21 and changes close to the significance level for miR-16 in humans in response to acute psychological social stress [23]. Intriguingly, the authors showed that the ΔCt-values of miR-21 were positively interrelated with miR-16, as also shown in our study.

This common correlation in the miR-16 and -21 expressions in two different species could suggest their potential orchestration role of the genes involved in the stress pathway response.

The analysis of the published literature to select salivary miRNAs to be tested in our study revealed numerous findings about miR-16 and miR-21, as related to stress-associated processes. MiR-16 and -21 were found to be involved in various biological events and, among these, in coping with oxidative stress [37], chronic mild stress [38], or psychological stress responses [39]. In Gerrard and collaborators’ research, both miRNA expressions were found up-regulated in the spinal cord tissue of rats affected by experimental autoimmune encephalomyelitis and subjected to a mild stress stimulus, such as the standing position [38]. Gidron earlier demonstrated that brief academic stress could alter the expression of miR-21 in blood. Interestingly, different levels of expression of miR-21 were found only in students with inadequate health behavior. The health-behavior score was attributed to physical exercise, diet, and stress control, which were tested before the trial [39]. The early literature identified and validated miR-16 as a miRNA targeting the serotonin transporter gene, and its expression correlated with the individual perception of stressors in humans [40,41]. Most of the studies converge in reporting that miR-16 targets genes in several stress-related processes. Zurawek and colleagues have demonstrated in a rodent model the correlations among the miR-16 level, state of depression, corticosteroid serum levels, and miRNA tissue levels in different brain areas [42]. Their results, according to other research, concluded that miR-16 may participate in the integrated stress responses when individuals are exposed to naturalistic stressors [32,43]. Due to the high homology among species of the miR-16 nucleotide sequence [44], it might be used for the genetic selection of ewes to identify resilient individuals [32,42,43].

In our study, the correlations between miRNAs and salivary cortisol levels were analyzed, but no significant results were found. Further insights may be gained from an analysis of the different phenotypes of the clustered animals given by measuring the behavioral patterns and temperament related to the stress response.

Indeed, it has been demonstrated that young ewes showing a calm temperament cope better with novel and potentially stressful situations, but breed differences exist [45].

## 5. Conclusions

Common breeding procedures related to management and handling, such as introducing a ram into the ewes’ flock, could represent stress stimuli for animals that show individual coping responses. Adequate control of these stressors is important to avoid the negative effects on animal health and production, but the selection of animals with more resilient tracts could help in this challenge [45].

At present, there is a gap in the literature about the involvement of miRNAs in the stress regulation mechanisms of sheep and, in particular, about circulating miRNAs. Further studies are needed to elucidate the specific roles of salivary miR-16 and miR-21 in sheep whose involvement in the stress pathway response was demonstrated in other species. The present study is limited to young, healthy individuals. Our results should be replicated in a large number of subjects of different breeds. However, the differential expression of these miRNAs between clusters of animals that show differences in cortisol concentration and trends attributes them to a potential role in the regulator of the stress response.

Salivary miRNAs might be potential biomarkers for the genetic selection of ewes to identify resilient individuals. Future studies should focus on examining the functional relevance of the miRNAs identified as resilience biomarkers in behavior and hormone patterns, with particular attention on the reproductive sphere to avoid the risk of selecting undesired phenotypes and reduce genetic variability in the population.

## Figures and Tables

**Figure 1 animals-13-03273-f001:**
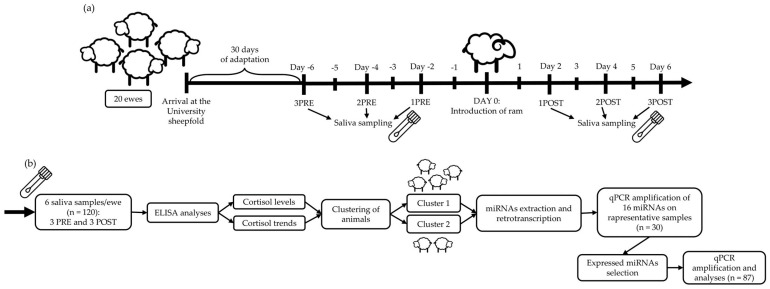
Overview of the experimental design. (**a**) Saliva was collected from 20 healthy sheep on alternate days 3 times before (PRE) and 3 after (POST) the introduction of a ram into the flock (6 samples/ewe). (**b**) Cortisol was extracted and two clusters of animals were identified. Saliva miRNAs were extracted, reverse-transcribed, and a panel of 16 miRNAs was amplified in a representative number of samples by qPCR. Specific expressed miRNAs were selected and amplified with qPCR in all samples of the two clusters. Then, statistical analyses were performed to detect DE-miRNAs. The figure is created by I. Manenti using PowerPoint 2021, Microsoft Corporation, Redmond, WA, USA. Icon designed by Freepik.

**Figure 2 animals-13-03273-f002:**
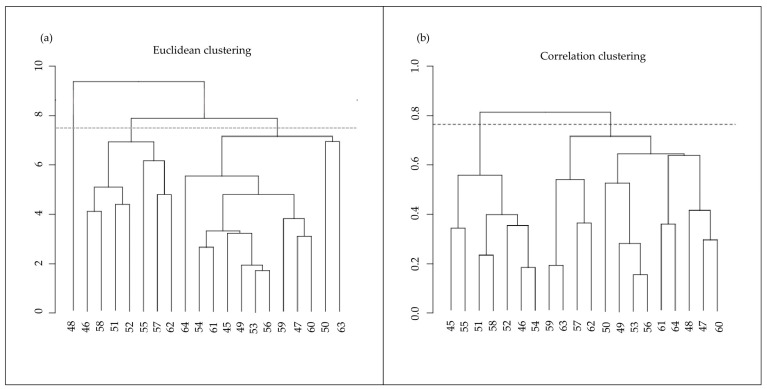
(**a**) Euclidean and (**b**) correlation clustering of animals based on salivary cortisol. Since according to Euclidean clustering (**a**), a sample clustered alone is not considerable, for the analyses based on Euclidean clustering, we focused on identifying three clusters to have two clusters comparable with those obtained from correlation-based clustering (**b**). By overlapping the similar subgroups obtained with the two measures, animals were divided into two clusters.

**Figure 3 animals-13-03273-f003:**
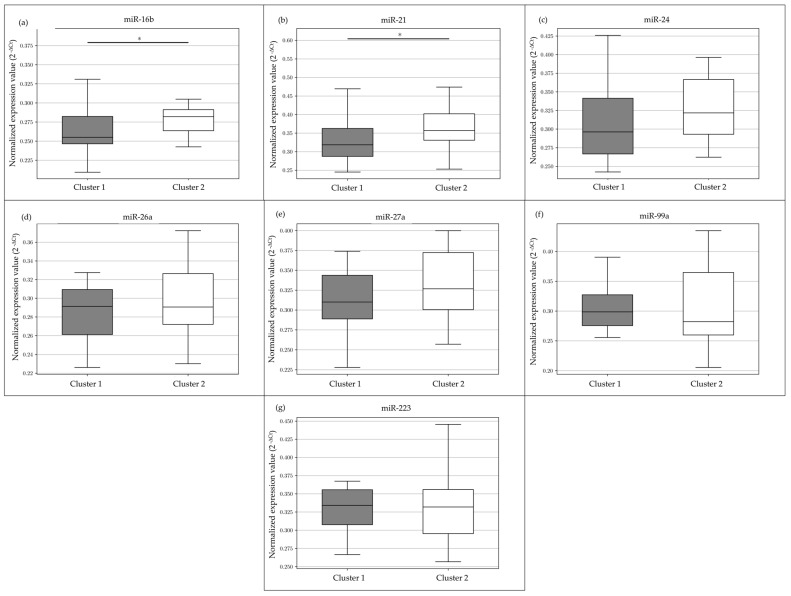
Expression of seven miRNAs selected (**a**–**g**). miR-16b (**a**) and miR-21 (**b**) show significant differences in expression between the two clusters. In the box plots, the thick central line represents the median; the top and bottom lines of the box represent the third quartile and the first quartile; whiskers indicate the variability in the data outside the third and first quartile. The figure shows the statistical significance of the overlap between the two groups determined using a Mann–Whitney U test; * *p* < 0.05.

**Figure 4 animals-13-03273-f004:**
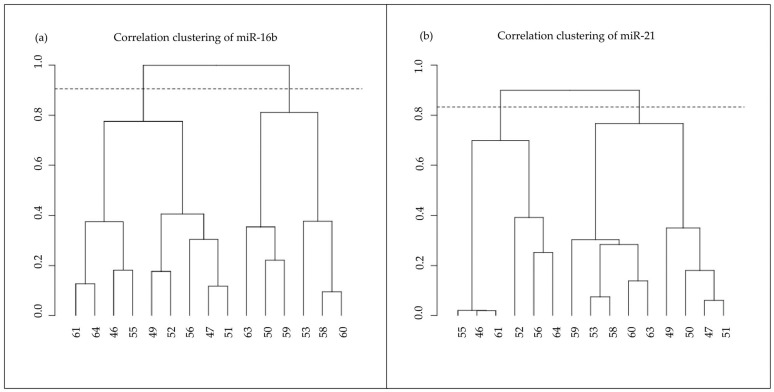
(**a**) miR-16b correlation clustering; (**b**) miR-21 correlation clustering. The similarity of the clusters was evaluated by the hypergeometric test based on a correlation distance between the two miRNAs’ expression profiles. The overlap of the most similar clusters based on miR-16b and miR-21 is significant (*p* = 0.017).

**Table 1 animals-13-03273-t001:** ID numbers of animals clustered and used for subsequent analysis: Cluster 1 consists of 10 animals, while Cluster 2 contains 5 animals.

Cluster 1	47 49 50 53 56 59 60 61 63 64
Cluster 2	46 51 52 55 58

**Table 2 animals-13-03273-t002:** MicroRNAs’ designation.

miRNA Name	Primer ID	Qiagen GeneGlobe ID
miR-223	hsa-miR-223-3p	YP00205986
miR-26a	hsa-miR-26a-5p	YP00206023
miR-24	hsa-miR-24-3p	YP00204260
miR-16b	oar-miR-16b	YP02115941
miR-191	cfa-miR-191	YP00205972
miR-27a	hsa-miR-27a-3p	YP00206038
miR-106a	oar-miR-106a	YP02110125
miR-17	oar-miR-17-5p	YP02110961
miR-29a	oar-miR-29a	YP02105177
miR-19b	hsa-miR-19b-3p	YP00204450
miR-30c	oar-miR-30c	YP02114922
miR-26b	hsa-miR-26b-5p	YP00204172
miR-23a	bta-miR-23a	YP02115608
miR-21	oar-miR-21	YP02110192
miR-221	bta-miR-221	YP02111922
miR-99a	bta- miR-99a-5p	YP00205945

## Data Availability

The data presented in this study are available upon request from the corresponding author. The data are not publicly available to preserve the privacy of the data.

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
