# Peer review of "Adaptation Response in Sheep: Ewes in Different Cortisol Clusters Reveal Changes in the Expression of Salivary miRNAs"

_animals, 2023, doi:10.3390/ani13203273_

Round 1
Reviewer 1 Report
The manuscript "Adaptation response in sheep: Ewes in different cortisol clusters reveal changes in the expression of salivary miRNAs" by Manenti et al aimed to determine the salivary miRNA profiles associated with 79 variable stress-responding in ewes following a common farm management practice: the introduction of a ram in the flock. In particular, consider that the ram was from the same ram, so that probably rams from a different farm could have improved the response.
The ms is well-written, the description of the methodology deep described, and the results well expressed. The discussion and conclusions are directly derived from the results. If some of my suggestions are included, the ms deserves to be published.
My main concern is about the stressor selected by the authors, which is the introduction of a ram in the flock. Please, include some previous references, if any, where cortisol levels or any other measurement of stress are reported after the introduction of the rams in the ewes' flock.
- Include live weight of the animals
- Is the ELISA Saliva Cortisol Kit validated for sheep samples? Please include references
- Figure 1 tries to present a chronology of the experiment design. Ram introduction is not included and a temporal series (with ram intro for instance as day 0) will help to understand the design.
- Line 154. SD or SE?
- Legends need more details of the experimental design
- Discussion: what about cortisol values before and after ram introduction? have they been compared?
I have nothing to comment
Reviewer 2 Report
Congratulations on the choice of topic and interesting findings. There were significant differences in cortisol levels among the sheep, which is a common occurrence. The reason for the rejection of 5 samples (10 samples in group 1 and only 5 in group 2) should be explained.
I believe it would be helpful to briefly present your perspective on the potential application of the results in practice in the discussion of results, especially in terms of maintaining variability in the population and whether selecting for stress, beyond the positive aspects, will have negative consequences.
Certainly, you can omit the general information about stress included in the introduction in the discussion of results.
